# Picosecond orientational dynamics of water in living cells

Martijn Tros[1], Linli Zheng[2], Johannes Hunger [3], Mischa Bonn[3], Daniel Bonn[4], Gertien J. Smits[2] & Sander Woutersen [1]

Cells are extremely crowded, and a central question in biology is how this affects the intracellular water. Here, we use ultrafast vibrational spectroscopy and dielectric-relaxation spectroscopy to observe the random orientational motion of water molecules inside living cells of three prototypical organisms: *Escherichia coli*, *Saccharomyces cerevisiae* (yeast), and spores of *Bacillus subtilis*. In all three organisms, most of the intracellular water exhibits the same random orientational motion as neat water (characteristic time constants ~9 and ~2 ps for the first-order and second-order orientational correlation functions), whereas a smaller fraction exhibits slower orientational dynamics. The fraction of slow intracellular water varies between organisms, ranging from ~20% in *E. coli* to ~45% in *B. subtilis* spores. Comparison with the water dynamics observed in solutions mimicking the chemical composition of (parts of) the cytosol shows that the slow water is bound mostly to proteins, and to a lesser extent to other biomolecules and ions.

[1] Van 't Hoff Institute for Molecular Sciences, University of Amsterdam, Science Park 904, 1098XH Amsterdam, The Netherlands. [2] Swammerdam Institute for Life Sciences, University of Amsterdam, Science Park 904, 1098XH Amsterdam, The Netherlands. [3] Max Planck Institute for Polymer Research, Department of Molecular spectroscopy, Ackermannweg 10, 55128 Mainz, Germany. [4] Institute of Physics, University of Amsterdam, Science Park 904, 1098XH Amsterdam, The Netherlands. Correspondence and requests for materials should be addressed to J.H. (email: hunger@mpip-mainz.mpg.de) or to G.J.S. (email: g.j.smits@uva.nl) or to S.W. (email: s.woutersen@uva.nl)

**W**ater plays a role in many cellular processes, ranging from protein folding to proton transport[1, 2]. Understanding the structure and dynamics of intracellular water is therefore important, but to what extent these properties differ from those of bulk water is still debated[3–9]. The high macroscopic viscosity of the cytoplasm (~$10^6$ times higher than water)[10] is mostly due to the presence of biomacromolecules. When the intracellular viscosity is probed using small particles or molecules, the observed viscosity decreases rapidly with the probe size[11, 12], with a sharp decrease below 50 nm, which can be regarded as the mesh size of the intracellular "gel"[13]. But even the smallest fluorescent probes still show intracellular rotational and translational diffusion times that are slower than in normal water[14–17], indicating that at the molecular level intracellular water is different from bulk water. Some of these differences stem from the different dynamics of water surrounding biomolecules[5, 9, 18–30], but the spatial extent of the effect of biomolecules on water dynamics remains debated. In the rather dilute aqueous solutions of specific biomolecules studied to date, their effect on the water stucture and dynamics is generally found to be short-ranged. However, such solutions are very different from crowded cells, where the high density of biomolecules might give rise to non-additive effects on the water dynamics.

Several methods have therefore been used to investigate the dynamics of cell water in vivo. Its low-frequency intermolecular vibrations have been studied using Kerr-effect[31] and THz[24, 32–36] spectroscopy. The orientational dynamics of cell water has been investigated using nuclear magnetic resonance (NMR):[37–40] from the frequency-dependent relaxation rate the distribution of rotational correlation times of the intracellular water can be determined, in particular for the water molecules exhibiting slow (>2 ns) dynamics. For water molecules exhibiting faster dynamics, averaged dynamical information can be obtained from NMR experiments[37–39]. The distribution of sub-ns reorientation times that underlie this average is difficult to access, and water molecules exhibiting picosecond orientational dynamics cannot be observed directly in NMR experiments. The rotation of such rapidly reorienting water molecules can be tracked in real time using ultrafast time-resolved infrared spectroscopy, which directly probes the random orientational motion of the water–OH bonds (or OD bonds in the case of deuterated water). This method has been used previously to investigate water dynamics in neat water[41–45], and aqueous solutions of salts[46–49] and biomolecules[20, 21, 50–52]. Alternatively, the collective orientational motion of the dipole moments of water molecules can be probed by measuring the electric-field induced polarization of a sample as function of field frequency using dielectric-relaxation spectroscopy (DRS)[53, 54].

Here, we combine these two spectroscopic methods to investigate the orientational dynamics of water in live cells of three prototypical species: a vegetatively growing bacterium (*Escherichia coli*) and a eukaryote (*Saccharomyces cerevisiae*, yeast) both living in aqueous environments, and *Bacillus subtilis* spores which can survive drought for many years and are resistant to heat, toxic chemicals and radiation. We find that in all three organisms most of the intracellular water exhibits the same random orientational motion as neat water, and that a smaller fraction (the magnitude of which varies between organisms) of the intracellular water exhibits slower orientational dynamics. Additional experiments in which we study the orientational water dynamics in solutions that mimic the cytosol or parts of it indicate that most of the slow intracellular water is bound to proteins.

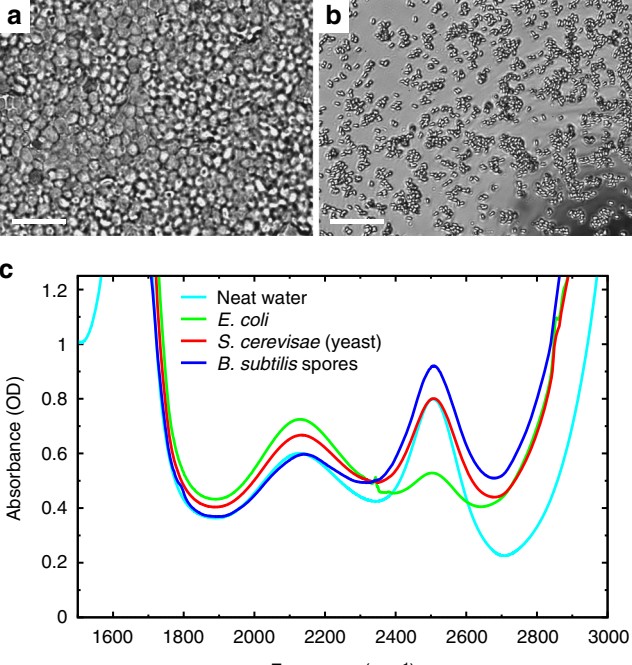

**Fig. 1** Organisms investigated in this study. Microscope images of the investigated samples of **a** *Saccharomyces cerevisiae* (yeast), and **b** *Bacillus subtilis* spores. The *bar* represents 10 μm. Note that in the yeast sample the extracellular space is filled with water, and in the spore sample mostly with air. **c** IR-absorption spectra of the samples. The peak at ~2500 cm$^{-1}$, due to the OD-stretch mode of HDO, is used to probe the orientational dynamics of the intracellular water

## Results

**Time-resolved vibrational spectrosopy.** In the time-resolved infrared experiments we use the intramolecular vibrations of water to probe its rotational dynamics. To avoid coupling between the molecular oscillators[55–57] we use the OD-stretch vibration (at ~2500 cm$^{-1}$) of isotopically diluted (HDO:$H_2O$) water. To ensure that the HDO:$H_2O$ isotope fraction is the same in the entire organism, the cells are grown or incubated in an aqueous environment containing isotopically diluted water (~5% HDO in $H_2O$). We find that the small deuterium fraction has no effect on the biological properties such as cell division and sporulation. In Fig. 1 we show IR spectra of the organisms investigated, together with microscope images of the samples. In all samples the amount of extracellular water was kept as low as possible (this issue will be further discussed below). During the experiments the samples are kept between two $CaF_2$ windows separated by a 25 μm teflon spacer. The OD-stretch mode of molecules other than water (such as sugars), and the ND-stretch mode (generated by NH/OD exchange) contribute negligibly to the absorbance (Supplementary Discussion 1). In all three organisms the OD-stretch absorption peak has the same shape and center frequency (to within 1 cm$^{-1}$) as that of bulk HDO:$H_2O$ water. The other main features in the IR spectra (at 1650, 2100, and >2800 cm$^{-1}$) are due to $H_2O$[58]. The small peak at ~2350 cm$^{-1}$ in the *E. coli* sample is due to $CO_2$[59], generated by the bacteria.

In the IR pump-probe experiments, a pump pulse preferentially excites ("tags") the stretching mode of OD bonds that are aligned along the IR polarization direction; the resulting anisotropic distribution of vibrationally excited OD groups is randomized by the random orientational motion of water molecules. This causes a decay in the anisotropy parameter

$R$, and, inversely, the anisotropy decay can be used to infer the reorientation dynamics of water molecules[43, 60]. Specifically, the OD-stretch anisotropy is proportional to the second-order correlation function of the orientation of the OD bonds of the HDO molecules: $R(t) = \frac{2}{5}\langle P_2(\mathbf{u}(0) \cdot \mathbf{u}(t))\rangle$, where $\mathbf{u}(t)$ is the unit vector along the OD bond and $P_2(x) = \frac{1}{2}(3x^2 - 1)$ the second-order Legendre polynomial, and where $\langle \dots \rangle$ denotes ensemble averaging. This function decays with increasing time $t$, and the decay mirrors the orientational memory loss due to the random orientational motion of the OD bonds. In Fig. 2 we compare the anisotropy decays of neat and intracellular water (the other OD-containing and ND-containing intracellular molecules contribute negligibly to the anisotropy decay, Supplementary Discussion 1). In neat water, the anisotropy decays exponentially to zero with a time constant $\tau_{or}^{IR} = 2.2$ ps, in agreement with previous studies[43]. The anisotropy of the cellular water decays with approximately the same time constant, but there is a residual anisotropy that persists up to our maximum accessible delay time of ~10 ps (determined by the lifetime of the vibrational excitation). We find that the decay of this residual anisotropy is too slow for its time constant to be determined from our measurements, and we can only conclude that it is larger than ~10 ps. These results imply that most of the water molecules exhibit orientational random motion on the same time scale as neat water, and a smaller fraction exhibits dynamics with $\tau_{or}^{IR}>2$ ps (in the remainder we will refer to this as "slow water"). This slow water fraction consists of water molecules with dynamics on a broad range of time scales (that cannot be distinguished in our experiment), and has been characterized in detail with NMR[37–40].

**Dielectric-relaxation spectroscopy.** We also investigate the dynamics of the cell water with dielectric-relaxation spectroscopy (DRS). With DRS we probe the polarization of a sample induced by an externally applied oscillating electric field. The response is measured as a frequency-dependent complex permittivity, with the real and imaginary parts representing the in-phase and out-of-phase (absorptive) components of the induced polarization[61]. For pure water (dashed lines in Fig. 3) the spectrum is dominated by a dispersion in the real permittivity and a corresponding peak in the imaginary permittivity at ~20 GHz. These signatures are characteristic for a relaxation mode, that is due to random orientational motion of the dipolar water molecules[62]. The 20 GHz frequency of the dielectric relaxation corresponds to a relaxation time of $\tau_{or}^{DRS} \sim 9$ ps (for neat water at 23 °C)[63]. This value of $\tau_{or}^{DRS}$ can be related to the orientational correlation time $\tau_{or}^{IR} \sim 2$ ps observed in the IR experiments by taking into account that in the IR experiments we measure the second-order orientational correlation time (see previous paragraph), whereas DRS is sensitive to the first-order orientational correlation function[64]. In particular, the DRS spectrum is determined by the correlation function $\langle \mathbf{P}(t) \cdot \mathbf{P}(0)\rangle$, where $\mathbf{P}(t)$ is the total polarization of the sample (arising mainly from the rotation of the water molecules, which have a permanent electric dipole moment), and $\langle \dots \rangle$ denotes ensemble averaging[61]. We find that the same 20 GHz relaxation also dominates the spectra of the organisms, with a somewhat reduced intensity (Fig. 3). Additionally, in cellular samples a low-amplitude relaxation is commonly observed at ~1 GHz (so-called $\delta$-relaxation)[65, 66]. This relaxation is not only due to the rotation of slowed-down water molecules but also to polarization of polyelectrolytes, rotation of low-molecular weight solutes, and conformational dynamics of proteins[65, 66].

To quantify the contribution of the dominant water relaxation we fit a combination of two Cole-Cole-type equations to the dielectric spectra (see Supplementary Table 3 for the fit parameters). These fits describe the spectra at frequencies ranging

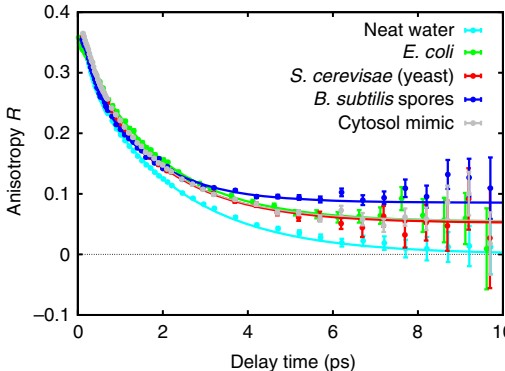

**Fig. 2** Time-resolved infrared spectroscopy measurements of water dynamics. Transient anisotropy of the OD-stretch mode (2508 cm$^{-1}$) of HDO water in different organisms and in a cytosol-mimic solution (pH 7, with protein). The *error bars* represent $1\sigma$. The *curves* are least-squares fits to exponential decays with a residual offset (see Supplementary Table 2 for the fit parameters)

from 760 MHz to 70 GHz very well (solid lines in Fig. 3). The contributions of the two relaxation processes to the imaginary permittivity are shown as shaded areas in Fig. 3. From this fit we find the orientational relaxation time $\tau_{or}^{DRS}$ of the dominant water relaxation in all three organisms to be very similar to the ~9 ps relaxation time of bulk water[63] (*E. coli*: $8.4 \pm 0.2$ ps, yeast: $9.1 \pm 0.2$ ps, spores: $8.9 \pm 0.1$ ps). Hence, most of the water in the investigated organisms exhibits picosecond dynamics that closely resembles the dynamics of neat water. The IR-anisotropy and DRS experiments thus give similar results, but it should be noted that these experiments probe different aspects of the water dynamics. Both experiments directly probe the random orientational motion of water molecules, but whereas the IR-anisotropy experiment probes the motion of individual water molecules, DRS probes the collective motion of all water molecules, and therefore is more sensitive to collective water dynamics. Hence, the IR results demonstrate that the local orientational dynamics of cell water is similar to that of neat water, and the DRS results show that the longer-ranged, collective dynamics of cellular water also does not differ significantly from that of neat water.

**Estimating the fractions of bulk-like and slow water.** The relative amounts of bulk-like and slow water in the different samples can be estimated from the relative amplitudes of the decay and the residual in the time-dependent IR anisotropy[50]. Combining this information with the water-mass fraction in the samples (obtained by drying the samples completely after the experiments, and comparing their mass before and after drying), we determine the mass fractions of bulk-like water, slow water, and dry mass in each organism (*blue*, *red*, and *green bars* in Fig. 4). Similarly, from the DRS data we can determine the volume fraction of bulk-like water simply by determining the reduction in amplitude of the bulk-like water-relaxation mode in the organisms as compared to neat water. As the relaxation strength is proportional to the volume fraction of the bulk-like water[61], these observed reductions of the amplitudes directly correspond to the volume fractions of bulk-like water of the organisms. They are indicated by the *gray bars* in Fig. 4. The bulk-like fractions obtained from the DRS measurements agree quite well with those obtained from the IR measurements. We note that the *error bars* in Fig. 4 should be regarded as lower estimates of the uncertainties in the numbers, since these *error bars* do not include the contribution of systematic

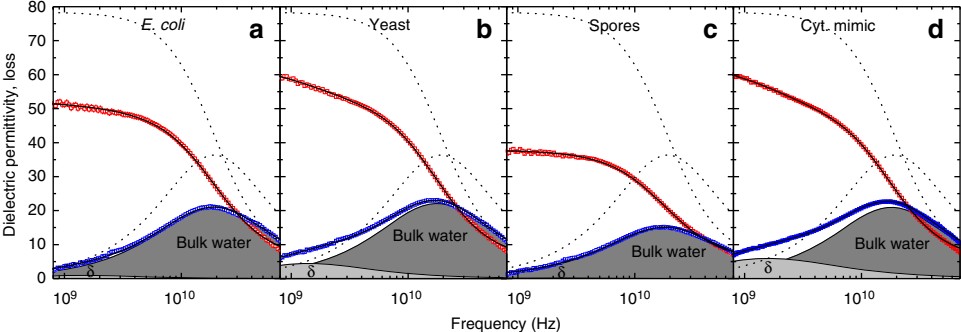

**Fig. 3** Dielectric-relaxation measurements of water dynamics. Real (*red points*) and imaginary (*blue points*) parts of the dielectric permittivity of three cellular samples and cytosol mimic (pH 7, with protein). The *dashed curves* show the permittivity of neat water. The *solid curves* are least-squares fits of a sum of two Cole–Cole modes to the data (see Supplementary Table 3 for the fit parameters). The *shaded areas* indicate the two contributions to the dielectric loss: bulk water and δ-relaxation. For clarity the Ohmic loss contribution (see Supplementary Discussion 2) has been subtracted

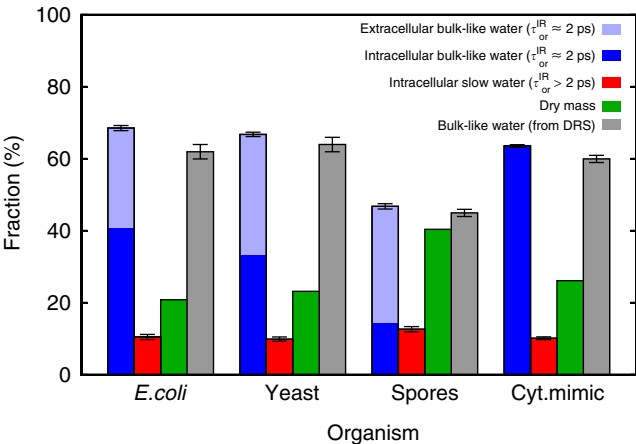

**Fig. 4** Fractions of bulk-like and slow water. From the data shown in Figs. 2 and 3 we can estimate the fractions of bulk-like and slow ($\tau_{or}^{IR} > 2$ ps) water in the different organisms and in the cytosol mimic (pH 7, with protein). The amount of extracellular water was determined by comparing the weight of the samples before and after complete drying and using the previously reported mass fractions of intracellular water in each of the organisms. The *error bars* represent 1σ, but they are lower limits of the actual uncertainties (see text)

errors and the effects of certain simplifying assumptions used in the data analysis (such as neglecting the kinetic-polarization and local-field effects in the DRS measurements, Supplementary Discussion 2, and the contribution of the NH-stretch and non-water–OH-stretch in the IR measurements), which are difficult to estimate precisely.

The bulk-like water fraction contains contributions from both intracellular and extracellular water. To determine the amount of extracellular water in our samples we proceed as follows. For all the investigated organisms the mass fraction of intracellular water with respect to the total cell mass is known (the intracellular water-mass fractions are ~70% for *E. coli*[67], ~65% for yeast[68], and ~40% for *B. subtilis* spores[69]). Using these known mass fractions, the total intracellular water mass present in each sample can be calculated from its dry weight. Subtracting this intracellular water mass from the total mass of water removed in the drying procedure yields the extracellular water mass in the sample. The uncertainties in the intracellular and extracellular water fractions obtained using this procedure are difficult to estimate, because the water content of a cell can vary depending on, e.g., the growth conditions, although it is generally maintained rather constant.

The extracellular part of the bulk-like fraction is indicated for the data obtained from the IR measurements by the light part of the *blue bars* in Fig. 4.

## Discussion

When comparing the fractions of intracellular bulk-like and slow water (*dark blue* and *red bars* in Fig. 4), we see that in all three organisms most of the intracellular water is bulk-like: ~80% of the water in *E. coli* and yeast, and ~55% of the water in the bacterial spores. The ratio of slow water to dry mass is roughly similar in all three organisms (0.5 for *E. coli*, 0.4 for yeast, 0.3 for the spores). This similarity suggests that the slow water is immobilized because it solvates biomolecules and/or ions (the difference in the ratios could then be due to differences in solvent-accessible surface area per unit of dry mass). To confirm this idea we measured the water dynamics of a solution which mimics yeast cytoplasm by having the same protein-mass fraction and ionic concentrations. The dielectric-relaxation spectrum of the cytosol mimic is very similar to that of the cells (Fig. 3); the same holds for the anisotropy decay (gray data points in Fig. 2) which shows a residual anisotropy just like the cells. The ratio of slow water to dry mass obtained from the residual anisotropy is also similar to those of the cells (Fig. 4). In contrast, in a solution containing only the ionic species of the cytosol mimic and no protein, the residual anisotropy is reduced by a factor of ~3 (see Supplementary Fig. 1). These findings indicate that a large part of the slow intracellular water is in the hydration shells of proteins or buried inside them (our measurements cannot distinguish these two types of slow water, but NMR shows that the buried fraction is very small[38]). Similar slowing down occurs in the hydration shells of other solutes such as osmolytes, DNA[21], and phospholipids[70]. The slowing down of the orientational dynamics of water molecules associated with ions and biomolecules is a well-known effect, and is mainly due to hydrogen bonding, electrostatics, and confinement effects[52, 71–75]. These same short-range interactions can cause templating of water by certain proteins[76] and charged planar membranes[77], an effect that is however limited to sub-nanometer length scales.

The most conspicuous difference between the water fractions in the different species is the lower fraction of bulk-like water in the spores. Bacterial spores can survive extreme conditions (heat, toxic chemicals, drought) for very long periods by effectively "shutting down" their biochemistry. To explain how this happens, it has been proposed that in the core of spores (which contains very little water) water might be in a glass-like state[78–80]. However, NMR measurements indicate that this is not

the case, and that the dormancy and heat resistance of spores probably result from protein immobilization[39, 40]. Our results confirm this idea, and show that most of the water in spores has exactly the same orientational mobility as bulk water: the relative amount of bulk-like water is less than in the other two organisms, but its dynamics is practically indistinguishable from that of bulk water. This might seem surprising in view of the extremely crowded intracellular matrix (with water constituting only ~40% of the spore-cell mass), which might suggest there cannot be any bulk-like water. A possible explanation could be that the water is inhomogeneously distributed in the spore cells[39], with "pockets" of bulk-like water. Such a water distribution would render the remainder of the cell extremely compact, and so increase its stability against chemical and thermal denaturation.

To conclude, most of the water inside living cells shows the same picosecond orientational dynamics as bulk neat water, both for individual water molecules (observed in the time-resolved IR experiments) and for collective reorientation (observed in the DRS experiments). Even in bacterial spores, which live under extreme conditions and contain only ~40% water, the majority of the intracellular water is indistinguishable from bulk water. In all organisms investigated there is a smaller fraction of water that reorients on time scales >2 ps (and that exhibits dynamics on many different time scales)[37–40], consisting mostly of water solvating biomolecules and ions, as evidenced by the similar fraction of slow water observed in a buffered protein solution mimicking the cytosol. Our results thus show that water inside living cells is mostly made up of "normal" (bulk-like) water, and partly of water water exhibiting slow orientational dynamics due to interaction with biomolecules and ions.

## Methods

**Preparation of the cells**. To obtain the cells a standard growing protocol for each of the three different organisms was followed. The *Bacillus subtilis* cells were grown overnight in a Tris-buffered saline (TBS) medium at a temperature of 37 °C. Subsequently, the cells were sporulated for 4 days in a 3-(N-morpholino)propa-nesulfonic (MOPS) medium containing water, $MgCl_2$, glucose, $NH_4Cl$, and tryptophan at 37 °C. To ensure that the spores for the infrared measurements contain ~5% HDO half of the culture was sporulated in a ~5% HDO medium. The other half of the cells was sporulated in a medium with 100% $H_2O$ and was used for the dielectric relaxation and and FTIR-background measurements. After 4 days of growing the spores were harvested and purified, thereby removing remaining non-sporulated cells from the sample. Finally, the samples were centrifuged for 30 min at 15,000 RPM, removing as much extracellular water as possible. The resulting cell pellet was used for the different measurements.

Cells of *S. cerevisiae* yeast were grown in a rich Yeast Peptone Dextrose (YPD) medium of 50 mL containing 20 g/L bacto peptone, 10 g/L yeast extract, and 20 g/L glucose (dextrose) overnight at 37 °C. *E. coli* was cultured in a 50 mL LB medium with 5 g/L yeast extract, 10 g/L bacto peptone, and 5 g/L NaCl. The media contained 100% $H_2O$, both cultures were grown overnight at a temperature of 37 °C. The ~5% HDO was added afterwards as there is free exchange of water through the cell membranes of these cells. After incubating the sample at room temperature interstitial water was removed by centrifuging for 30 min at 15,000 RPM.

**Cytosol-mimic preparation**. Four different solutions mimicking the cytosol were prepared using a procedure reported previously[81]. The concentrations of different chemical species in the solutions are listed in Supplementary Table 1. The pH was set either to 5.0 or 7.0, and for each pH value two solutions were made, one with bovine serum albumin (BSA) protein and one without. BSA is a globular and highly soluble protein from blood plasm that is often used as a model protein, in particular in studies concerning molecular crowding in the cell. The protein concentration in the cytosol mimic was approximately 30%.

**Experiments**. Images of the spores and yeast cells were recorded using an Olympus IX71 wide field light microscope. Fourier-transform infrared (FTIR) measurements were performed on a PerkinElmer Spectrum Two spectrometer with a resolution of 0.5 cm$^{-1}$. Fourier-transform infrared spectra of the cells and cytosol mimics were recorded using IR sample cells consisting of two 2 mm thick CaF2 windows separated by a 25 μm teflon spacer. These sample cells were also used in the time-resolved measurements. Additional spectra of the cells and cytosol mimics in 100% $H_2O$ were recorded for background subtraction. Polarization-

resolved infrared pump-probe experiments were done using a setup described previously[74]. Pump-beam scattering by the samples was eliminated by delaying every second pump pulse by half an optical cycle using a photo-elastic modulator and averaging the signals at these two pump-probe delays[82]. The thermal contribution to the pump-probe signals was taken into account in the data analysis using a procedure described previously[60].

Complex dielectric spectra of the samples were recorded in the frequency range from 0.76 to 70 GHz with a coaxial reflectometer based on an Anritsu Vector Star MS4647A vector network analyser with an open ended coaxial probe based on 1.85 mm coaxial connectors[83–85]. To calibrate the instrument for directivity, frequency response, and source-match errors we use water[62], air, and conductive silver paste as calibration standards[83]. The organisms were measured by putting small amounts of the sample on the probe head until the response plateaued upon addition of additional sample volume. The spectra of the cytosol mimic were measured by immersing the probe in the solution. Error bars were obtained from at least three reproduced experiments.

**Data availability**. The data that support the findings of this study are available from the corresponding authors on reasonable request.

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

## Acknowledgements
We would like to thank Michiel Hilbers for taking the microscope pictures.

## Author contributions
G.S., D.B., M.B., and S.W. conceived the experiments, L.Z. and M.T. prepared the samples, M.T. carried out the experiments, S.W. supervised the vibrational-spectroscopy experiments, and data analysis. J.H. supervised the dielectric-relaxation experiments and data analysis, G.S. supervised the biological and biochemical procedures, and analysis. All authors contributed to writing the manuscript.

## Additional information

**Competing interests:** The authors declare no competing financial interests.

