## [Peer Review File · Nature Communications]

Reviewers' comments:

Reviewer #1 (Remarks to the Author):

In this manuscript, Woutersen and coworkers have shown that vibrational and dielectric relaxation spectroscopies are powerful techniques to clearly distinguish between two kinds of water (bulk-like fast and biomolecules-associated slow water) in living cells in terms of their orientational dynamics. This is important because of many indispensable roles of water dynamics in different cellular processes.

There have been quite a few ultrafast spectroscopic studies on water confined in different model systems such as lipid planar membranes, reverse micelles, and gyroid lipid/surfactant structures, but this study is the first one (to the best of my knowledge) that actually studied the effects of three different actual cellular environments on water dynamical behavior. It is unique, novel, and has a broad scientific appeal, and therefore it is suitable for Nature Communication. However, there are a few issues that must be addressed and the manuscript needs to be revised before publication. These issues are listed as follows:

(a) The main conclusions of the manuscripts are based on the determination of the first- and second-order orientational correlation functions. However, nowhere in the manuscript those functions were defined or introduced clearly. This study is aimed for a broad scientific community, therefore the authors should explain what these correlation functions physically mean in 3-5 sentences.

(b) The anisotropy decay in Fig. 2 has clearly a bi-exponential behavior for all cases: The fast decay ($t < 2$ ps) is identical for all cases, indicating the presence of bulk-like water. The interesting and important part is the slow decay ($t > 2$ ps) that distinguishes between the cellular systems studied here. I would recommend that either the authors make a bi-exponential fits to these curves or a single exponential fit after $t > 2$ ps for more accurate estimation of orientational dynamical time scales.

(c) The authors assigned the water molecules that exhibit slow orientational dynamics to those that are associated with biomolecules and ions, but did not discuss the molecular origin/driving forces responsible for this. The previous studies on the aforementioned model systems showed that it's mainly due to hydrogen bonding, electrostatics, and confinement effects (The Journal of Chemical Physics 137, 094706 (2012), The Journal of Chemical Physics 141, 22D505 (2014), J. Am. Chem. Soc., 2016, 138 (8), pp 2472–2475, J. Phys. Chem. C, 2007, 111 (25), pp 8884–8891).

Woutersen and coworkers are expert vibrational spectroscopists and it should be trivial for them to measure two-dimensional infrared/sum-frequency generation spectra of water as a function of waiting time (t_2), which should reveal the molecular origin of such slow dynamic water. Without these measurements, this current study appears to be incomplete.

(d) The authors should be careful in making the statement in the conclusion that, water in cells are actually 'normal water' which is an overstatement. I would rather stick to the statement that most water in cells behave like 'normal water', but they become slow dynamic in the close proximity of biomolecules such as proteins and ionic species.

Reviewer #2 (Remarks to the Author):

In their manuscript Tros et al. convincingly quantify for the first time the amounts of bound and free

water in living cells. Water is here classified as bound when its reorientation takes longer than the 10 ps timescale for the second-order orientational correlation function covered by ultrafast vibrational spectroscopy. The thus obtained fractions of bound water, which differ significantly between active cells and spores, are in reasonable agreement with the outcome of the performed dielectric relaxation experiments, probing the amount of free (pure-water-like) water. Most important, the authors clearly show that the dynamics of the intra-cellular free water is practically identical to that of neat water despite large solute crowding in the cell. Thus, the effect of biomolecules on surrounding water molecules is only short-ranged –affecting the bound water. For the remaining free water there is no long-range order imposed by the solute (proteins, nucleic acids etc.), in contrast to claims in the literature. To use the catchy title of Ref. 5: The authors show that there is water in biology but no biological water.

The present manuscript is suitable for publication in Nature Communications after the corrections & additions indicated below.

1. Page 4, lines 1ff: With the comment Ref. 55 the authors refer to the Supporting Information for experimental details but for many points addressed here the details are only given in the Methods section. The authors should also direct the interested reader to that section.
2. The dielectric relaxation times of free water reported for the cells but also for the cytosol mimic (Table S3) are larger than that of pure water (8.3 ps) and the difference exceeds the quoted error. Has this any significance or might it be a temperature effect as the present data are at 23 °C whereas the pure water value is for 25 °C (at 20 °C pure water has 9.4 ps)?
3. The relaxation at ~1 GHz is certainly not only due to slow water but interfacial polarization effects at the cellular interfaces are almost certainly at lower frequencies. More likely are dipolar low-molecular weight solutes in the cytosol and intramolecular flexibility of proteins etc.
4. The spectrum of what cytosol mimic is shown in Fig. 3? Is this with or without added protein? The figure caption does not tell this.
5. The total amounts of bulk-like water detected by ultrafast vibrational spectroscopy and by DRS are indeed in reasonable agreement. Nevertheless there seem to be significant deviations for E. coli and the cytosol mimic (again: which one; also Table S3) judging from the error bars. Can the authors comment on that?
6. Page 9, Experimental details (lines 3-7): The authors write “The cytosol-mimic solutions were kept in an IR cell consisting of two 2-mm thick CaF2 windows ...” and in the next sentence “Fourier-transform ... and cytosol mimics were recorded using IR sample cells consisting of two 1-mm thick CaF2 windows ...” So, which windows were used now?
7. On page 10 a frequency range of 76 MHz to 70 GHz is claimed for the dielectric spectra but on page 9 of the Supplementary Information and in Fig. 3 it is only 0.76 GHz to 70 GHz.
8. Supplementary Information, page 2, line 3: Definitely, Ref. 60 of the article does not tell how cytosol mimics should be prepared.
9. 1st paragraph of page 4: “See the following pages for the details” certainly does not mean page 5.

Reviewer #3 (Remarks to the Author):

The study by Tros employs a combination of time-resolved vibrational spectroscopy and dielectric relaxation spectroscopy to demonstrate that most of the water in cells behaves like normal bulk water, while a small fraction at the vicinity of solutes (proteins, ions, etc.) is modestly slowed down. Per se, this result is not new since it has been obtained by several techniques (e.g., NMR) already decades ago. However, given the recent hype about “cellular water” when numerous researchers (to some extent including several of the authors of the present study) brought up unsubstantiated claims of the existence of protein-induced long-range ordering of water or in some cases even a special “gell-like” water in living matter, the present ms. brings a refreshing air of common sense. For this contextual

reason I believe it would be a service to the community if this work, which seems to be technically competently executed, is published. Nevertheless, the authors should first consider the following issues:

1. The whole interpretation of the present measurements relies on dividing the water content in the cell into two categories - normal and slow. This is clearly a simplification and the present conclusions are at least to some extent victims of this approximation. This should be thoroughly discussed in the ms. so that the reader gets a clear idea to what extent the presented conclusions are dependent on the above assumption.

1. p. 2, 1st par.: Please drop emotional statements like "Water, the solvent of life..." from the ms. and spare them for future PR. Also, when you talk about "the high macroscopic intracellular viscosity..." you should say explicitly of what.

2. p. 5, top (and elsewhere in the ms.: Use ">" instead of ">>" when comparing relaxation times of the slow and normal water. I would spare ">>" for the previous claims of orders of magnitude slow down, here it is only a factor of five or so.

3. p. 7: The text makes the impression that the only solutes that slow down water in cells are proteins and ions. How about non-ionic osmolytes, DNA, phospholipids, etc.?

4. In the concluding section it would be fair if the authors put the present results in context of their earlier claims concerning protein-induced long-range effects on water (e.g., in *Sci. Adv.* 2016, 2, e1501778).

REVIEWERS' COMMENTS:

Reviewer #1 (Remarks to the Author):

The authors carefully and successfully addressed all the questions in the response letter. The revised manuscript looks great and should be accepted for publication.

Reviewer #2 (Remarks to the Author):

The authors satisfactorily addressed all points of my previous review. I am happy now with the manuscript as it is.

Reviewer #3 (Remarks to the Author):

I think the author made a fair effort to accommodate the comments by the reviewers, thus the manuscript may be published in my opinion.

Reply to the reviews of MS# NCOMMS-17-09606, “Picosecond orientational dynamics of water in living cells”

We kindly thank all three reviewers for their positive and constructive comments on our manuscript, and are happy that they all recommend its publication in Nature Communications.

In the revision of the text, we have taken the reviewers’ comments and suggestions into account, and we believe this has resulted in a more convincing and clearer manuscript. Our answers to the reviewers’ comments are given below, in the same order as they appear in their reviews. In the resubmission we have included a version of the manuscript in which all modified and added text is highlighted in red. In our reply, the reviewer comments are printed in italics and our answers in normal font.

Reviewer #1

In this manuscript, Woutersen and coworkers have shown that vibrational and dielectric relaxation spectroscopies are powerful techniques to clearly distinguish between two kinds of water (bulk-like fast and biomolecules-associated slow water) in living cells in terms of their orientational dynamics. This is important because of many indispensable roles of water dynamics in different cellular processes.

There have been quite a few ultrafast spectroscopic studies on water confined in different model systems such as lipid planner membranes, reverse micelles, and gyroid lipid/surfactant structures, but this study is the first one (to the best of my knowledge) that actually studied the effects of three different actual cellular environments on water dynamical behavior. It is unique, novel, and has a broad scientific appeal, and therefore it is suitable for Nature Communication. However, there are a few issues that must be addressed and the manuscript needs to be revised before publication. These issues are listed as follows:

(a) The main conclusions of the manuscripts are based on the determination of the first- and second-order orientational correlation functions. However, nowhere in the manuscript those functions were defined or introduced clearly. This study is aimed for a broad scientific community, therefore the authors should explain what these correlation functions physically mean in 3-5 sentences.

AUTHOR REPLY We thank the reviewer for this suggestion, which we have taken to heart.

CHANGE TO TEXT Added definitions and short explanations of the orientational correlation functions (p. 4 and 5).

(b) The anisotropy decay in Fig. 2 has clearly a bi-exponential behavior for all cases: The fast decay ($t < 2$ ps) is identical for all cases, indicating the presence of bulk-like water. The interesting and important part is the slow decay ($t > 2$ ps) that distinguishes between the cellular systems studied here. I would recommend that either the authors make a bi-exponential fits to these curves or a single exponential fit after $t > 2$ ps for more accurate estimation of orientational dynamical time scales.

AUTHOR REPLY We thank the reviewer for this suggestion, and have tried to fit bi-exponential decays to our data. In these fits we did not include a constant residual (as we did in Fig. 2), but only two exponential decays. Unfortunately, due to the (inevitably) large error bars on the data points at long delay times, the uncertainties in the

slow rate constants obtained from these least-squares fits were actually larger than the small ($<0.1 \text{ ps}^{-1}$) values of these slow rate constants themselves. As such, this analysis proposed by the reviewer reveals that, while we cannot quantify the slow relaxation time, we can set a lower limit on the slow reorientation time of $\sim 10 \text{ ps}$. This is a relevant conclusion, which we present in the revised manuscript.

CHANGE TO TEXT Added statement summarizing the above (p. 5).

(c) The authors assigned the water molecules that exhibit slow orientational dynamics to those that are associated with biomolecules and ions, but did not discuss the molecular origin/driving forces responsible for this. The previous studies on the aforementioned model systems showed that it's mainly due to hydrogen bonding, electrostatics, and confinement effects (The Journal of Chemical Physics 137, 094706 (2012), The Journal of Chemical Physics 141, 22D505 (2014), J. Am. Chem. Soc., 2016, 138 (8), pp 2472–2475, J. Phys. Chem. C, 2007, 111 (25), pp 8884–8891). Woutersen and coworkers are expert vibrational spectroscopists and it should be trivial for them to measure two-dimensional infrared/sum-frequency generation spectra of water as a function of waiting time (t_2), which should reveal the molecular origin of such slow dynamic water. Without these measurements, this current study appears to be incomplete.

AUTHOR REPLY We completely agree with the reviewer that our manuscript would benefit from a discussion regarding the origin of the slow orientational dynamics. Time-dependent 2DIR spectra of the biological samples would shed light on the OD-stretch frequency fluctuations of the water molecules (which mirror the hydrogen-bond fluctuations), and could shed light on possible contributions to the slowdown of water molecules. However, the execution of those experiments on these (light-scattering) biological samples is highly nontrivial and would furthermore be beyond the scope of the present manuscript: our work is focused on the orientational dynamics, and we use these dynamics to quantify the population of two distinct sub-ensembles of water in biological systems. Attempts to further disentangle one of those two sub-ensembles (i.e. that of the slow water molecules) into contributions from different types of water molecules is certainly interesting, but constitutes a major research effort of itself, especially for all of the different systems reported in our manuscript. In any case, we have included a discussion about the different contributions for the slow water fraction in the revised manuscript, referring also to the previous work suggested by the reviewer.

CHANGE TO TEXT Added brief discussion and references explaining possible origins of the slow orientational dynamics of solvating water molecules (p. 8, line 8-10).

(d) The authors should be careful in making the statement in the conclusion that, water in cells are actually 'normal water' which is an overstatement. I would rather stick to the statement that most water in cells behave like 'normal water', but they become slow dynamic in the close proximity of biomolecules such as proteins and ionic species.

AUTHOR REPLY We thank the reviewer for this excellent suggestion, which we have taken to heart.

CHANGE TO TEXT Formulated conclusion more precisely, as proposed by the reviewer.

Reviewer #2

In their manuscript Tros et al. convincingly quantify for the first time the amounts of bound and free water in living cells. Water is here classified as bound when its reorientation takes longer than the 10 ps timescale for the second-order orientational correlation function covered by ultrafast vibrational spectroscopy. The thus obtained fractions of bound water, which differ significantly between active cells and spores, are in reasonable agreement with the outcome of the performed dielectric relaxation experiments, probing the amount of free (pure-water-like) water. Most important, the authors clearly show that the dynamics of the intra-cellular free water is practically identical to that of neat water despite large solute crowding in the cell. Thus, the effect of biomolecules on surrounding water molecules is only short-ranged –affecting the bound water. For the remaining free water there is no long-range order imposed by the solute (proteins, nucleic acids etc.), in contrast to claims in the literature. To use the catchy title of Ref. 5: The authors show that there is water in biology but no biological water.

The present manuscript is suitable for publication in Nature Communications after the corrections & additions indicated below.

1. Page 4, lines 1ff: With the comment Ref. 55 the authors refer to the Supporting Information for experimental details but for many points addressed here the details are only given in the Methods section. The authors should also direct the interested reader to that section.

AUTHOR REPLY AND CHANGE TO TEXT We thank the referee for noting this point, and have changed the comment in Ref. 55 accordingly.

2. The dielectric relaxation times of free water reported for the cells but also for the cytosol mimic (Table S3) are larger than that of pure water (8.3 ps) and the difference exceeds the quoted error. Has this any significance or might it be a temperature effect as the present data are at 23 °C whereas the pure water value is for 25 °C (at 20 °C pure water has 9.4 ps)?

AUTHOR REPLY AND CHANGE TO TEXT The reviewer is correct that the value of 8.3 ps referred to in our original submission is the relaxation time of neat water at 25°C, while our experiments were performed at 23°C. The somewhat higher values for the relaxation times for the cells stem almost certainly from the experimental temperature of 23°C (the interpolated relaxation time for pure water at 23°C is ~9 ps). To avoid confusion, we refer throughout the revised manuscript to the more appropriate value of ~9 ps for neat water at 23°C, with a reference to *Chem. Phys. Lett.* **306** 1999 57–63.

3. The relaxation at ~1 GHz is certainly not only due to slow water but interfacial polarization effects at the cellular interfaces are almost certainly at lower frequencies. More likely are dipolar low-molecular weight solutes in the cytosol and intramolecular flexibility of proteins etc.

We thank the reviewer for this comment, and we agree that cells are too large objects to observe significant interfacial polarizations at frequencies as high as 1 GHz (where a relaxation commonly referred to as delta-relaxation is observed, see also Ref. 65). Thus, besides polarization of polyelectrolytes, low-molecular weight solutes and proteins may indeed contribute. We have revised the manuscript accordingly and have

included a new reference [*Biochim. Biophys. Acta* **1810** (2011), 727–740] which summarizes recent literature that discusses the origins of this delta-relaxation.

4. *The spectrum of what cytosol mimic is shown in Fig. 3? Is this with or without added protein? The figure caption does not tell this.*

AUTHOR REPLY AND CHANGE TO TEXT The spectrum shown is that of the cytosol mimic including protein, at pH 7. In the revised text we state this in the caption of Fig. 3.

5. *The total amounts of bulk-like water detected by ultrafast vibrational spectroscopy and by DRS are indeed in reasonable agreement. Nevertheless there seem to be significant deviations for E. coli and the cytosol mimic (again: which one; also Table S3) judging from the error bars. Can the authors comment on that?*

AUTHOR REPLY We are hesitant to speculate about the difference between the bulk-water fractions as obtained from the IR and dielectric-relaxation experiments. The problem is that even though these differences are somewhat larger than the error bars on the data (as the reviewer correctly remarks), these error bars (obtained by averaging several measurements) do not take into account systematic errors (such as kinetic depolarization and local-field effects in the DRS experiments [discussed in the SI] and the small contribution of NH- and non-water-OH groups to the OH-stretch absorption in the IR experiments), which are unfortunately difficult to estimate. We therefore prefer to point out in the revised manuscript that the error bars in Fig. 4 should be taken as lower estimates as they do not take systematic errors into account, and that the differences between the bulk fractions are probably within the uncertainty of the experiments.

CHANGE TO TEXT Added the above considerations to the discussion (p. 7); specified the cytosol mimic used (in the caption of Fig. 4 and in Table S3).

6. *Page 9, Experimental details (lines 3-7): The authors write “The cytosol-mimic solutions were kept in an IR cell consisting of two 2-mm thick CaF2 windows ...” and in the next sentence “Fourier-transform ... and cytosol mimics were recorded using IR sample cells consisting of two 1-mm thick CaF2 windows ...” So, which windows were used now?*

AUTHOR REPLY AND CHANGE TO TEXT We apologize for the confusion. We used 2 mm thick windows, and have corrected the typo in the revised manuscript.

7. *On page 10 a frequency range of 76 MHz to 70 GHz is claimed for the dielectric spectra but on page 9 of the Supplementary Information and in Fig. 3 it is only 0.76 GHz to 70 GHz.*

AUTHOR REPLY AND CHANGE TO TEXT We thank the reviewer for pointing out this oversight to us. The 76 MHz number was a typo (should have read 0.76 GHz or 760 MHz), which we have corrected in the revised manuscript.

8. *Supplementary Information, page 2, line 3: Definitely, Ref. 60 of the article does not tell how cytosol mimics should be prepared.*

AUTHOR REPLY AND CHANGE TO TEXT We kindly acknowledge the reviewer for spotting this typo. In the revised text we refer to the correct article describing the preparation of the cytosol mimic.

9. 1st paragraph of page 4: "See the following pages for the details" certainly does not mean page 5.

AUTHOR REPLY AND CHANGE TO TEXT We thank the reviewer for spotting this typo, which we have corrected in the revised manuscript.

Reviewer #3

The study by Tros employs a combination of time-resolved vibrational spectroscopy and dielectric relaxation spectroscopy to demonstrate that most of the water in cells behaves like normal bulk water, while a small fraction at the vicinity of solutes (proteins, ions, etc.) is modestly slowed down. Per se, this result is not new since it has been obtained by several techniques (e.g., NMR) already decades ago. However, given the recent hype about "cellular water" when numerous researchers (to some extent including several of the authors of the present study) brought up unsubstantiated claims of the existence of protein-induced long-range ordering of water or in some cases even a special "gel-like" water in living matter, the present ms. brings a refreshing air of common sense. For this contextual reason I believe it would be a service to the community if this work, which seems to be technically competently executed, is published. Nevertheless, the authors should first consider the following issues:

1. The whole interpretation of the present measurements relies on dividing the water content in the cell into two categories - normal and slow. This is clearly a simplification and the present conclusions are at least to some extent victims of this approximation. This should be thoroughly discussed in the ms. so that the reader gets a clear idea to what extent the presented conclusions are dependent on the above assumption.

AUTHOR REPLY We agree completely with the reviewer, and have improved our discussion accordingly. As discussed in our reply to point (b) of reviewer 1, the most specific quantitative statement we can make about the slow category is that it exhibits orientational dynamics on time scales longer than ~10 ps. The slow category thus includes a broad range of time scales, of which the >25 ps part has previously been sampled by NMR.

CHANGE TO TEXT We have extended the discussion by stating explicitly that the slow component is a mixture of different water species with reorientation times longer than ~10 ps (p. 5, line 10-12 and page 9, lines 14-15).

1. p. 2, 1st par.: Please drop emotional statements like "Water, the solvent of life..." from the ms. and spare them for future PR. Also, when you talk about "the high macroscopic intracellular viscosity..." you should say explicitly of what.

AUTHOR REPLY AND CHANGE TO TEXT We appreciate the reviewer's advice and have toned down our phrasing. In the revised manuscript we also specify that we refer to the viscosity of the cytoplasm (p. 2)

2. p. 5, top (and elsewhere in the ms.: Use ">" instead of ">>" when comparing relaxation times of the slow and normal water. I would spare ">>" for the previous claims of orders of magnitude slow down, here it is only a factor of five or so.

AUTHOR REPLY AND CHANGE TO TEXT We agree with the reviewer, and we have modified the text accordingly.

3. p. 7: *The text makes the impression that the only solutes that slow down water in cells are proteins and ions. How about non-ionic osmolytes, DNA, phospholipids, etc.?*

AUTHOR REPLY AND CHANGE TO TEXT We agree with the reviewer, and on p. 8 we have therefore added a sentence about these other solutes that also slow down the water, with references to Elsaesser's work on water hydrating DNA and phospholipids. We thank the reviewer for pointing out this issue to us.

4. *In the concluding section it would be fair if the authors put the present results in context of their earlier claims concerning protein-induced long-range effects on water (e.g., in *Sci. Adv.* 2016, 2, e1501778).*

AUTHOR REPLY We assume that the reviewer is referring to our paper "Ice-nucleating bacteria control the order and dynamics of interfacial water", *Sci. Adv.* **2**, e1501630, 2016 (rather than the one mentioned by the reviewer, which concerns strain-induced changes in the secondary structure of the protein fibrin). In that paper it is stated that: "...stronger binding of water in the Thr region might affect water order in neighboring regions by long-range interactions within the water network or improved rigidity throughout the side-chain lattice"; and also that: "Molecular alignment within the H-bonding network of water can promote long-range energetic coupling and therefore, by effectively funneling heat away from the interface, promote the formation of critical ice embryos necessary for nucleation." We did not quantify "long-range" in that manuscript, which is indeed a rather serious omission; we intended to communicate that the length scale is long compared to the intermolecular length scales, but certainly did not intend to imply nanometers. Indeed, the MD simulations show that the orientational correlation function of water molecules decays on sub-nanometer length scales; this sub-nanometer length scale is very typical, even for highly (positively or negatively) charged planar membranes, which are expected to achieve maximum water templating, see e.g. Roy *et al.*, *J. Chem. Phys.* **141**, 18C502 (2014). We would like to note that it only takes a limited number of coupled O-H groups to delocalize excess vibrational energy, as observed for the ice nucleating protein.

CHANGE TO TEXT We have added (on p. 8) the following brief discussion to the revised manuscript, referring to the papers discussed above: "Short-range interactions can cause templating of water by certain proteins [*Sci. Adv.* **2**, e1501630 (2016)] and charged planar membranes [*J. Chem. Phys.* **141**, 18C502 (2014)], an effect that is however limited to sub-nanometer length scales."